# Inflammation and Insulin Resistance in Diabetic Chronic Coronary Syndrome Patients

**DOI:** 10.3390/nu15122808

**Published:** 2023-06-19

**Authors:** Tianyu Li, Peizhi Wang, Xiaozeng Wang, Zhenyu Liu, Zheng Zhang, Yongzhen Zhang, Zhifang Wang, Yingqing Feng, Qingsheng Wang, Xiaogang Guo, Xiaofang Tang, Jingjing Xu, Ying Song, Yan Chen, Na Xu, Yi Yao, Ru Liu, Pei Zhu, Yaling Han, Jinqing Yuan

**Affiliations:** 1National Clinical Research Center for Cardiovascular Diseases, State Key Laboratory of Cardiovascular Disease, Fuwai Hospital, National Center for Cardiovascular Diseases, Chinese Academy of Medical Sciences and Peking Union Medical College, Beijing 100037, China; tyli14@fudan.edu.cn (T.L.); wangpeizhi_fuwai@126.com (P.W.); 2Cardiovascular Research Institute, Department of Cardiology, General Hospital of Northern Theater Command, Shenyang 110016, China; wxiaozeng@163.com (X.W.); hanyaling@263.net (Y.H.); 3Department of Cardiology, Peking Union Medical College Hospital, Chinese Academy of Medical Sciences and Peking Union Medical College, Beijing 100730, China; pumch_lzy@163.com; 4Department of Cardiology, The First Hospital of Lanzhou University, Lanzhou 730000, China; zhangccu@163.com; 5Department of Cardiology, Peking University Third Hospital, Beijing 100191, China; zhangy_zhen@163.com; 6Department of Cardiology, Xinxiang Central Hospital, Xinxiang 453002, China; wzf2146@163.com; 7Department of Cardiology, Guangdong Cardiovascular Institute, Guangzhou 510100, China; fyq1819@163.com; 8Department of Cardiology, The First Hospital of Qinhuangdao, Qinhuangdao 066000, China; qswang1960@yahoo.com.cn; 9Department of Cardiology, The First Affiliated Hospital of Zhejiang University, Hangzhou 314400, China; gxg22222@zju.edu.cn; 10Department of Cardiology, Fuwai Hospital, National Center for Cardiovascular Diseases, Chinese Academy of Medical Sciences and Peking Union Medical College, Beijing 100037, China; tangxiaoxiaotang@126.com (X.T.); jjxu1984@aliyun.com (J.X.); jessiesong1983@aliyun.com (Y.S.); chen_yan04@163.com (Y.C.); smilexuna@163.com (N.X.); yaoyi0118@163.com (Y.Y.); liuru@fuwaihospital.org (R.L.); drzhupei@126.com (P.Z.)

**Keywords:** triglyceride–glucose index, insulin resistance, high-sensitivity C-reactive protein, inflammation, diabetes, chronic coronary syndrome, cardiac event, mediation analysis

## Abstract

Limited evidence exists on the combined and mediating effects of systemic inflammation on the association between insulin resistance and cardiovascular events in patients with diabetes and chronic coronary syndrome (CCS). This secondary analysis of a multicenter prospective cohort included 4419 diabetic CCS patients. Triglyceride–glucose index (TyG) and high-sensitivity C-reactive protein (hsCRP) were applied to evaluate insulin resistance and systemic inflammation, respectively. The primary endpoint was major adverse cardiac event (MACE). Associations of TyG and hsCRP with cardiovascular events were estimated using Cox regression. A mediation analysis was performed to assess whether hsCRP mediates the relationship between TyG and cardiovascular events. Within a median 2.1-year follow-up period, 405 MACEs occurred. Patients with high levels of TyG and hsCRP experienced the highest MACE risk (hazard ratio = 1.82, 95% confidence interval: 1.24–2.70, *p* = 0.002) compared to individuals with low levels of both markers. HsCRP significantly mediated 14.37% of the relationship between TyG and MACE (*p* < 0.001). In diabetic CCS patients, insulin resistance and systemic inflammation synergically increased the risk of cardiovascular events, and systemic inflammation partially mediated the association between insulin resistance and clinical outcomes. Combining TyG and hsCRP can help identify high-risk patients. Controlling inflammation in patients with insulin resistance may bring added benefits.

## 1. Introduction

Coronary artery disease (CAD) reigns as the foremost single cause of mortality and the loss of disability-adjusted life years across low-income, middle-income, and high-income countries [1]. Diabetes stands as the primary complication of CAD, with cardiovascular causes emerging as the leading contributors to mortality among diabetic patients [2]. By 2020, the prevalence of diabetes among Chinese adults had reached 12.8%, affecting a population of 130 million individuals [3]. Globally, it is estimated that the number of diabetic patients will be 700 million by 2045 [4]. Despite the advances in secondary prevention strategies and invasive treatment techniques, patients with comorbid traits continue to face a high risk of recurrent cardiovascular events [5,6,7]. Therefore, identifying individuals with modifiable nontraditional cardiovascular risk factors is crucial to implementing effective comprehensive risk reduction and improving their prognosis.

Insulin resistance is the metabolic characteristic of type 2 diabetes and a common manifestation of type 1 diabetes. Insulin resistance contributes to the development of atherosclerosis by causing glucose and lipid metabolism disorder, endothelial dysfunction, coagulation disorders, and smooth muscle cell dysfunction [8], and it has been established as an independent cardiovascular risk factor in diabetic patients [9,10,11]. Triglyceride–glucose index (TyG), derived from fasting triglyceride and blood glucose levels, has emerged as a reliable measure of insulin resistance. It offers the advantages of being readily available, inexpensive, and independent of insulin treatment status. Recent research supports that TyG is independently associated with cardiovascular outcomes in diabetic and non-diabetic patients with different CAD phenotypes [8]. However, the underlying mechanism remains unclear.

As a chronic condition associated with systemic low-grade inflammation, diabetes promotes vascular inflammation by upregulating the expression of pro-inflammatory genes [12]. Inflammation is widely acknowledged as an etiological factor in atherosclerosis and has emerged as another well-established independent cardiovascular risk factor in general and diabetic patients [13,14,15]. High-sensitivity C reactive protein (hsCRP) is a widely used biomarker of systemic inflammation, and its predictive significance for cardiovascular events has been extensively validated in various CAD settings [14,15,16,17].

Both insulin resistance and inflammation play causal roles in atherosclerosis. Their proatherogenic processes share some cells and cytokines but also involve different signaling pathways. Moreover, insulin resistance and inflammation initiate and aggravate each other in a vicious cycle [18]. Additionally, insulin resistance exacerbates atherosclerosis by causing glucose and lipid metabolism disorders and endothelial and smooth muscle dysfunction. These processes often involve the release of pro-inflammatory cytokines [8]. The aforementioned biological findings have given rise to two hypotheses for this study. Firstly, the combined effect of insulin resistance, as measured by TyG, and systemic inflammation, as assessed by hsCRP, may worsen cardiovascular outcomes. Secondly, hsCRP may serve as a mediator between TyG and cardiovascular events.

This study aimed to investigate the combined and mediating effects of hsCRP-reflected inflammation on the association between TyG-reflected insulin resistance and cardiovascular events in diabetic patients with comorbid chronic coronary syndrome (CCS). For this paper, patients were first categorized into four groups based on high and low levels of TyG and hsCRP to evaluate the combined effect of TyG and hsCRP on multiple cardiovascular events. Subsequently, after assessing the prerequisites for mediation analysis, a mediation analysis was performed on eligible events to examine the role of hsCRP in the impact of TyG on these events.

## 2. Materials and Methods

### 2.1. Study Design and Participants

The present study utilized data from the PRospective Observational Multi-center cohort for ISchemic and hEmorrhage risk in coronary artery disease patients (PROMISE), designed to develop ischemic and bleeding risk scores specifically for Chinese populations. The PROMISE study recruited 18,701 hospitalized CAD patients from nine centers across China between January 2015 and May 2019. Inclusion criteria included patients of at least 18 years old, diagnosis of CAD, indication of at least one antiplatelet drug, and willingness to participate in the study by providing informed consent. Exclusion criteria were a life expectancy of fewer than six months and current participation in another interventional clinical trial. The decision to initiate invasive treatment was made by the heart team according to current guidelines, taking into account the patient’s preferences. Guideline-recommended secondary prevention medications were prescribed for all patients without documented contraindications. The PROMISE study complied with the Declaration of Helsinki and was approved by the Ethics Committee of Fuwai Hospital (protocol code: 2013-449, date of approval: 4 September 2013; protocol code: No. 2017-860, date of approval: 10 January 2017). All participants provided written informed consent.

The participant selection process for the present study is outlined in Figure 1. Patients diagnosed with diabetes were included. Patients with acute conditions that may lead to stress hyperglycemia or alter systemic inflammation status, including acute coronary syndrome, active infection or tuberculosis, systemic inflammatory diseases, and malignancy, were excluded from this study. Participants without fasting blood glucose, triglyceride, and hsCRP values and those not having aspirin or statins at baseline were also excluded. A total of 4419 patients were eligible for this study, of whom 1228 (27.79%) were female. They were divided into four groups based on TyG and hsCRP levels: low TyG and hsCRP (L-TyG/L-hsCRP), low TyG with high hsCRP (L-TyG/H-hsCRP), high TyG with low hsCRP (H-TyG/L-hsCRP), and high TyG and hsCRP (H-TyG/H-hsCRP).

### 2.2. Measures of Insulin Resistance and Systemic Inflammation

Insulin resistance was evaluated using TyG, calculated as Ln (fasting triglyceride (mg/dL) × fasting blood glucose (mg/dL) ÷ 2) [19], with triglyceride converted from mmol/L to mg/dL by multiplying by 88.50. Systemic inflammation was evaluated using hsCRP. Higher TyG and hsCRP levels indicate increased insulin resistance and systemic inflammation, respectively. TyG was categorized at the optimal cut-off value of 8.46. HsCRP was categorized at 2.00 mg/L, aligning with the definition of residual inflammatory risk [20].

### 2.3. Blood Sampling and Laboratory Testing

Fasting blood samples were drawn in the morning as part of routine clinical practice. Blood glucose was assayed using an enzymatic hexokinase method. Glycated hemoglobin (HbA1c) was measured with automated glycohemoglobin analyzers (Tosoh HLC-723G8, Tokyo, Japan). Lipid profiles were measured with automatic biochemistry analyzers (Hitachi 7150, Tokyo, Japan). HsCRP was measured by rate turbidimetry with immunoassay analyzers (Beckman Assay, Brea, CA, USA).

### 2.4. Endpoints and Follow-Up

The primary endpoint was major adverse cardiac event (MACE), a composite of cardiac death, non-fatal myocardial infarction, and any revascularization. Secondary endpoints included all-cause death and each component of the composite endpoint. Deaths without an unequivocal non-cardiac cause were presumed to be cardiac. Myocardial infarction was diagnosed following the Fourth Universal Definition of Myocardial Infarction. Any revascularization was defined as percutaneous coronary intervention (PCI) or coronary artery bypass grafting for any lesion driven by ischemic symptoms or events. Outcome data and medication usage were followed up through clinic visits, telephone calls, text messages, and letters by an independent group of clinical research coordinators one and two years after discharge. Two independent cardiologists adjudicated endpoint events, and disagreements were resolved through consensus.

### 2.5. Definition of Variables

Diabetes was defined as fasting blood glucose ≥7.0 mmol/L (126.00 mg/dL), HbA1c ≥ 6.50% (48 mmol/mol), 2 h blood glucose of oral glucose tolerance test ≥11.1 mmol/L, oral antidiabetic medication or insulin use, or self-reported diabetes. Chronic kidney disease was defined as an estimated glomerular filtration rate < 60 mL/min/1.73 m^2^ or self-reported chronic kidney disease. Hypertension was defined as a blood pressure ≥ 140/90 mmHg on repeated measurements, antihypertensive medication use, or self-reported history of hypertension. Dyslipidemia was defined when at least one of the following criteria was met: total cholesterol ≥ 6.22 mmol/L, triglyceride ≥ 2.26 mmol/L, low-density lipoprotein cholesterol ≥ 4.14 mmol/L, high-density lipoprotein cholesterol < 1.04 mmol/L, lipid-lowering medication use, or self-reported history of dyslipidemia. Three-vessel disease was defined as angiographically significant stenosis (≥50%) of all three main coronary arteries. Postprocedural Thrombolysis in Myocardial Infarction (TIMI) flow grade 3 was considered successful PCI; otherwise, it was considered unsuccessful. The medication adherence for aspirin and statins was categorized as follows: 2 years of regular use, >1 to <2 years of regular use, and <1 year of regular use or irregular use.

### 2.6. Statistical Analysis

Given the low rate of missing data in the study dataset, missing values were imputed using the median or mode as appropriate for the variable type (Appendix A). The correlation between TyG and hsCRP was assessed using Spearman’s rank correlation analysis. Categorical variables were expressed as numbers (percentages) and compared using the χ^2^ test. Non-normally distributed continuous variables (by the Kolmogorov–Smirnov test) were expressed as medians (interquartile ranges) and compared using the Kruskal–Wallis H test.

Kaplan–Meier survival curves were compared using the log-rank test. Associations of insulin resistance and/or systemic inflammation with cardiovascular events were reported as hazard ratios (HRs) and 95% confidence intervals (CIs) estimated using Cox proportional-hazards regression. The optimal cut-off value for TyG was determined using the maximum Youden index of the receiver operating characteristic curve for predicting MACE. Time-dependent receiver operating characteristic analysis was performed using the survivalROC Package. Restricted cubic splines with 4 knots were fitted using the rms Package when TyG and hsCRP were analyzed continuously. The knot locations were set at the 5th, 35th, 65th, and 95th percentiles of TyG and hsCRP levels.

Covariables for adjustment included age (continuous), sex, body mass index (BMI) (continuous), smoking history, insulin use, peripheral artery disease, left ventricular ejection fraction <40%, left main stem/three-vessel disease, synergy between percutaneous coronary intervention with Taxus and cardiac surgery (SYNTAX) score (categorical), PCI status, aspirin adherence, and statins adherence.

Pre-specified subgroup analyses were performed based on five variables: sex, age (≥65 years vs. <65 years, according to the definition of elderly for the Chinese populations), BMI (≥28 kg/m^2^ vs. <28 kg/m^2^, according to the definition of obesity for the Chinese populations), insulin use, and PCI status. Three pre-specified sensitivity analyses were performed to examine the robustness of the results: dividing patients into nine groups by employing cut-off values of 8.46 and 9.31 for TyG and 1.00 and 3.00 mg/L for hsCRP, excluding patients who lost to 2-year follow-up, and excluding those with hsCRP > 10 mg/L. One post-hoc sensitivity analysis was performed by excluding patients with imputed values.

A causal mediation analysis was performed using the mediation Package developed by Imai et al. [21] based on the counterfactual-based framework to assess whether systemic inflammation mediates the association between insulin resistance and cardiovascular events. A directed acyclic graph was used to visualize the assumed causal model, with TyG (continuous) as the exposure, hsCRP (continuous) as the mediator, and endpoint events as the outcome variables. Confounders identified through the directed acyclic graph were adjusted, including age, sex, BMI, smoking history, insulin use, hypertension, dyslipidemia, chronic kidney disease, left ventricular ejection fraction < 40%, HbA1c, aspirin adherence, and statin adherence. Given the time-to-event nature of the outcomes, the Cox proportional-hazards regression model was chosen for mediation analysis. The significance of the mediating effect was examined through 1000 bootstrap samples.

Statistical analyses were conducted with R version 4.2.0 (R Core Team 2022, Vienna, Austria). Two-tailed *p* values < 0.05 were considered statistically significant. Figures were prepared with GraphPad Prism version 9.0.0 (GraphPad Software, San Diego, CA, USA).

## 3. Results

### 3.1. Study Population and Baseline Characteristics

Of the 4419 participants, the median age was 62 years (interquartile range: 55–68 years), and 1228 (27.79%) were female. Insulin use was reported in 635 (14.37%) patients. Nearly two-thirds of the participants underwent PCI during hospitalization, with a success rate of 95.46%. The median TyG value was 8.96 (interquartile range: 8.59–9.38), while the median hsCRP value was 1.46 mg/L (interquartile range: 0.70–2.92 mg/L).

Table 1 shows baseline characteristics across patient groups categorized by TyG and hsCRP levels. The percentage of patients with high hsCRP levels was higher in the H-TyG group (37.61%) than in the L-TyG group (25.50%). The proportions of female, obese patients, current smokers, and patients undergoing PCI increased sequentially from the L-TyG/L-hsCRP group to the H-TyG/H-hsCRP group, as did the levels of HbA1c and low-density lipoprotein cholesterol. Regardless of their TyG levels, patients with H-hsCRP were more likely to present a higher lesion severity, as reflected by the presence of left main/three-vessel disease and a higher SYNTAX score. Insulin use, comorbidities other than dyslipidemia, reduced cardiac function, and medication adherence were comparable among the four groups.

### 3.2. Association of TyG and hsCRP with Cardiovascular Events

Of the total 4419 participants, 4362 (98.71%) completed a median follow-up of 2.1 years (interquartile range: 2.0–2.3 years), with 405 MACEs, 85 all-cause deaths, 51 cardiac deaths, 118 myocardial infarctions, and 282 cases of any revascularization occurring. The MACE-free survival curves illustrate that patients in the H-TyG/H-hsCRP group had the highest incidence of MACE. Patients with isolated high TyG or high hsCRP levels also experienced more MACE than those in the L-TyG/L-hsCRP group (*p* for trend = 0.001, Figure 2A). Event-free survival curves for secondary endpoints revealed similar trends (all *p* for trend < 0.05, Figure 2B–E).

Univariate Cox analysis for clinical outcomes is shown in Appendix A. Cox analyses before and after adjustment consistently yielded that the H-TyG/H-hsCRP group had a significantly higher MACE risk compared to the L-TyG/L-hsCRP group (adjusted HR: 1.83, 95% CI: 1.24–2.70, *p* = 0.002), followed by the H-TyG/L-hsCRP group (adjusted HR: 1.78, 95% CI: 1.22–2.60, *p* = 0.003). However, the MACE risk did not significantly increase for the L-TyG/H-hsCRP group. The risks of all secondary endpoints were also significantly higher in patients with high levels of both TyG and hsCRP than in those with low levels of both (Table 2).

When patients were grouped solely by TyG or hsCRP levels, high levels of either biomarker were still significantly associated with a higher MACE risk. However, there was no significant difference in the risk of any revascularization between the H-hsCRP and L-hsCRP groups. Additionally, no interaction was observed between TyG and hsCRP categories (Appendix A). The relationship between cardiovascular events and TyG or hsCRP on a continuous scale is plotted in Appendix A.

### 3.3. Subgroup Analyses and Sensitivity Analyses

Subgroup analyses by sex and PCI status yielded consistent results with the primary analysis that the H-TyG/H-hsCRP group had the highest MACE risk, followed by the H-TyG/L-hsCRP group. However, it should be noted that the small sample size of the unsuccessful PCI subgroup (*n* = 129) may have limited the statistical power. Similar results were only observed in non-elderly, non-obese, and insulin-naïve patients, whereas none of the groups in their counterpart subpopulations were significantly associated with MACE risk. Nevertheless, no significant interactions were detected in the subgroup analysis for any subgroup variables, except for BMI (Figure 3).

Sensitivity analyses supported the robustness of the main results (Appendix A). When patients were divided into nine groups, the MACE risk gradually increased across the low, moderate, and high TyG categories for each given hsCRP level. Likewise, the MACE risk also increased as hsCRP levels increased for each given TyG category. Patients in the H-TyG/H-hsCRP group experienced the highest MACE risk. Robust results in agreement with the primary analysis were observed after excluding 57 patients who did not complete the 2-year follow-up, 229 with very high hsCRP levels (>10 mg/L), or 197 with imputed values.

### 3.4. Mediation Analysis

According to the prerequisites for conducting a mediation analysis, TyG should be correlated with hsCRP, both TyG and hsCRP should be associated with cardiovascular events, and the association between TyG and cardiovascular events should be attenuated when adjusting for hsCRP. Appendix A shows a weak but significant correlation between TyG and hsCRP (Spearman correlation r = 0.196, *p* < 0.001). When analyzed as continuous variables, the risk of any revascularization increased with elevated TyG rather than hsCRP levels, whereas the risk of myocardial infarction increased with elevated hsCRP but not TyG levels. Both TyG and hsCRP levels were associated with the risks of MACE, all-cause death, and cardiac death regardless of adjustment, with the estimates of HRs of TyG being reduced when additionally adjusted for hsCRP (Appendix A). Therefore, mediation analysis was restricted to MACE, all-cause death, and cardiac death. The analysis demonstrated a significant partial mediating effect of systemic inflammation on the association of insulin resistance with MACE, all-cause death, and cardiac death after adjustment for measured confounding. Specifically, hsCRP was found to mediate 14.37% of the association between TyG and MACE, 17.70% of the association between TyG and all-cause death, and 15.57% of the association between TyG and cardiac death. TyG remained directly associated with MACE, all-cause death, and cardiac death (Figure 4).

## 4. Discussion

This study showed that the combination of TyG-measured insulin resistance and hsCRP-measured systemic inflammation significantly increased the 2-year risk of multiple cardiovascular events in diabetic CCS patients, including MACE, all-cause death, cardiac death, myocardial infarction, and any revascularization. HsCRP partially mediated the impact of TyG on MACE, all-cause death, and cardiac death.

Insulin resistance causes hyperglycemia and hypertriglyceridemia, enhances oxidative stress and vascular inflammation, induces endothelial dysfunction characterized by a pro-inflammatory, pro-thrombotic, vasoconstrictive phenotype, and promotes smooth muscle cell proliferation and collagen deposition, which lead to atherosclerosis and cardiovascular events [8]. Tao et al. demonstrated the predictive value of TyG for cardiovascular events [8]. This study confirmed a significant association between TyG-assessed insulin resistance and MACE in diabetic patients with CCS. Likewise, this study also revealed that hsCRP-assessed inflammation was an independent cardiovascular risk factor for diabetic patients with CCS. Both insulin resistance and systemic inflammation are metabolic characteristics of diabetes and independent risk factors for cardiovascular events. However, few epidemiological studies have investigated the causal pathways through which these factors contribute to cardiovascular events.

This study found that high TyG and hsCRP levels identified individuals at the highest cardiovascular risk, with significantly increased risks of MACE and secondary outcome events compared with those with lower levels of both markers. These findings may be attributed to the combined effects of insulin resistance and inflammation in promoting atherosclerosis. Insulin resistance and inflammation contribute strongly to endothelial dysfunction [8,10,13], and hyperglycemia can also damage the endothelium, albeit to a lesser extent [22]. HsCRP appears to have the potential to damage the endothelium directly as well [16,17,23]. Then, increased triglyceride-rich lipoproteins pass through the compromised endothelial barrier [24]. In an insulin-resistant and hyperglycemic environment, oxidized or glycated lipoproteins trigger a heightened inflammatory response beneath the endothelium [22], ultimately exacerbating atherosclerosis and increasing plaque instability. Furthermore, insulin resistance and systemic inflammation mutually amplify each other in a vicious cycle [10,18]. Tissues may become more sensitive to pro-inflammatory stimuli in the presence of hyperinsulinemia [18].

Another finding of this study was that hsCRP partially mediated the impact of TyG on MACE, all-cause death, and cardiac death in diabetic patients with CCS. Similarly, a previous study revealed that hsCRP partially mediated the association between insulin resistance and poor clinical outcomes in non-diabetic patients with ischemic stroke [25]. These observations provide epidemiological evidence supporting the biologically plausible hypothesis that inflammation plays a role in the link between insulin resistance and cardiovascular events. Insulin resistance and hyperglycemia can activate the NLRP3 (NOD (nucleotide oligomerization domain)-, LRR (leucine-rich repeat)-, and PYD (pyrin domain)-containing protein 3) inflammasome, leading to the release of cytokines such as interleukin-1, interleukin-6, and tumor necrosis factor-α. The NLRP3 pathway is a critical element in atherosclerotic pathogenesis, and hsCRP is a downstream marker of these cytokines. Therefore, elevated TyG levels can exacerbate atherosclerosis through an upregulated NLRP3 signaling pathway represented by hsCRP [8,10,22]. However, the link between TyG and cardiovascular events may involve other mediators, such as traditional cardiovascular risks (hypertension and dyslipidemia) and unidentified factors. In addition, hsCRP does not capture other pro-atherogenic inflammatory pathways, such as Toll-like receptors, proprotein convertase subtilisin/kexin type 9, Notch, and Wnt pathways [13]. Further studies are required to explore the mechanisms underlying the causal relationship between TyG and cardiovascular events.

Due to the lower likelihood of females being admitted to PCI-capable hospitals than males [26], the proportion of females in this study may be slightly lower than the actual representation in the CAD population. However, this does not impact the finding of a consistent combined effect of insulin resistance and inflammation on cardiovascular outcomes in both sexes. In elderly and insulin-using patients, the prognostic significance of TyG/hsCRP levels was not observed. However, no significant interaction between TyG/hsCRP levels and age or insulin use was detected, indicating that this observation may be attributed to the small sample sizes and low statistical power of these subpopulations. A meta-analysis has demonstrated that TyG stably predicts mortality in the elderly [27], and its predictive ability is not affected by insulin treatment status [28]. In addition, the observed interaction between BMI and TyG/hsCRP levels was likely due to chance and cannot be interpreted as evidence of the “obesity paradox”, as the median hsCRP value was lower in obese patients with hsCRP ≥ 2.00 mg/L than non-obese patients, despite a higher proportion of individuals with hsCRP ≥ 2.00 mg/L in the obese group than the non-obese group.

Clinical trials have demonstrated the potential of anti-inflammatory therapy for improving cardiovascular outcomes in the secondary prevention of CAD. However, there are still challenges in incorporating anti-inflammatory therapy into CAD management. One hurdle is the need for more precise risk stratification to determine patients who would benefit the most, thereby improving cost-effectiveness. This study indicates that combining TyG and hsCRP can help identify patients with extremely high cardiovascular risk. Anti-inflammatory therapy for these individuals may provide additional benefits beyond directly attenuating systemic inflammation, including reducing the synergistic and mediating effects of inflammation on the negative prognostic impact of insulin resistance.

The limitations of this study should be acknowledged. Firstly, while potential confounders were identified and adjusted, unknown or unmeasured confounding cannot be ruled out due to the observational nature of the study. For instance, we were unable to adjust for the types and dosages of statins and aspirin, both of which are known to have anti-inflammatory effects. Secondly, the study cohort primarily consisted of individuals of Han Chinese ethnicity. Caution should be exercised when generalizing the results to other racial and ethical populations. Thirdly, the results of this study may be affected by potential regression dilution bias and temporal changes in TyG and hsCRP levels, given that these biomarkers were only measured once at baseline. Baseline hsCRP values can be influenced by several non-pathological factors, such as age, sex, obesity, hormone replacement therapy, and smoking. Relying on a single measurement of hsCRP may lead to an inaccurate assessment of inflammation. Lastly, since TyG and hsCRP measurements were collected simultaneously, the logical time order between the exposure and the mediator cannot be assured.

## 5. Conclusions

In diabetic CCS patients, TyG and hsCRP synergically increased the 2-year risk of multiple cardiovascular events, including MACE, all-cause death, cardiac death, myocardial infarction, and any revascularization. HsCRP partially mediated the impact of TyG on MACE, all-cause death, and cardiac death. Combining insulin resistance and systemic inflammation can help identify high-risk patients, and controlling inflammation in patients with insulin resistance may bring added benefits. The study was limited by potential unmeasured confounding, limited generalizability to other ethnic populations, single-time measurement of biomarkers, and the inability to establish the temporal order between the exposure and the mediator, which should be considered when interpreting the findings.

## Figures and Tables

**Figure 1 nutrients-15-02808-f001:**
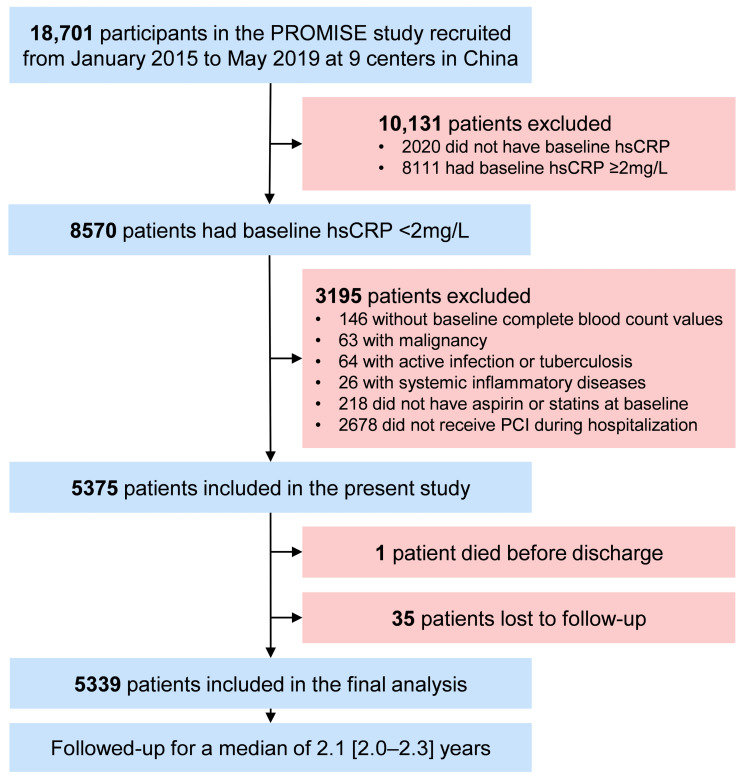
Study flowchart. Abbreviations: FBG, fasting blood glucose; H, high; hsCRP, high-sensitivity C-reactive protein; L, low; PROMISE, the PRospective Observational Multi-center cohort for ISchemic and hEmorrhage risk in coronary artery disease patients; TyG, triglyceride–glucose index.

**Figure 2 nutrients-15-02808-f002:**
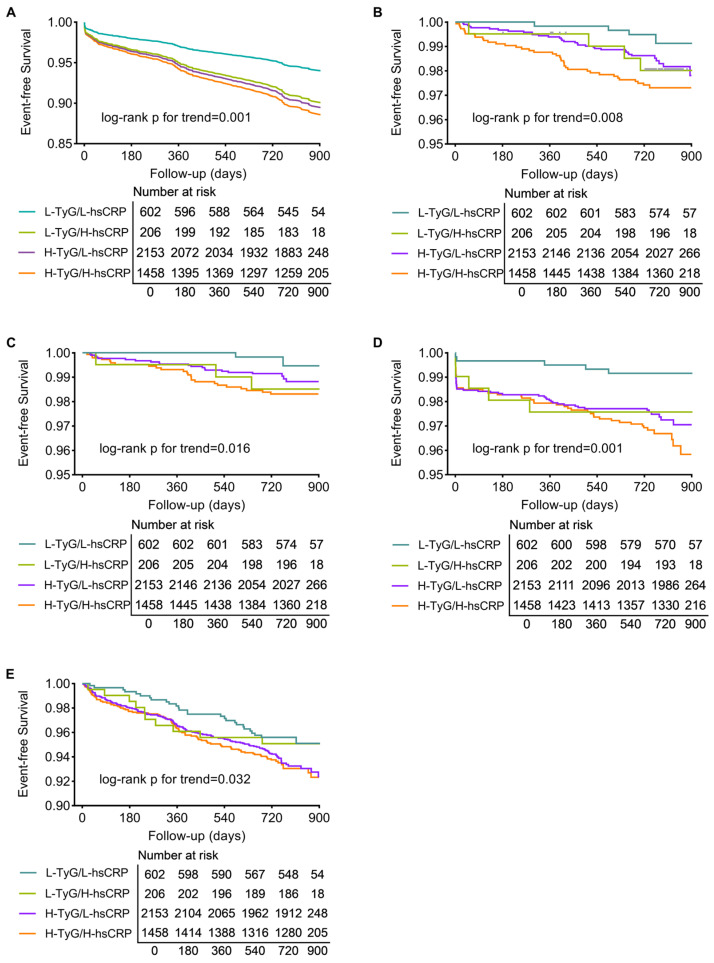
Kaplan–Meier analysis of patients grouped by TyG and hsCRP levels. Survival curves for MACE (**A**), all-cause death (**B**), cardiac death (**C**), myocardial infarction (**D**), and any revascularization (**E**). Abbreviations: H, high; hsCRP, high-sensitivity C-reactive protein; L, low; MACE, major adverse cardiac event; TyG, triglyceride–glucose index.

**Figure 3 nutrients-15-02808-f003:**
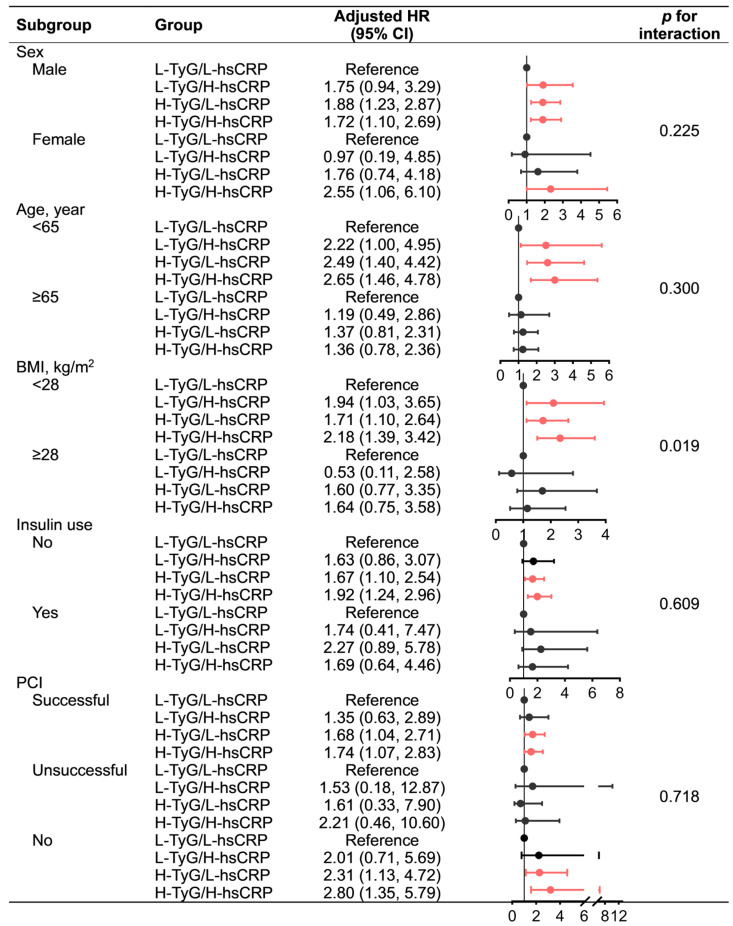
Subgroup analysis for the association of TyG and hsCRP with MACE. Red indicates statistical significance, while black indicates statistical non-significance. Adjusted for age (continuous), sex, BMI (continuous), smoking history, peripheral artery disease, chronic kidney disease, prior myocardial infarction, prior stroke, prior revascularization, low-density lipoprotein cholesterol ≤ 1.8 mmol/L, left ventricular ejection fraction < 40%, left main stem/three-vessel disease, synergy between percutaneous coronary intervention with Taxus and cardiac surgery score (categorical), percutaneous coronary intervention status, aspirin adherence, statin adherence. Abbreviations: BMI, body mass index; CI, confidence interval; H, high; HR, hazard ratio; hsCRP, high-sensitivity C-reactive protein; L, low; MACE, major adverse cardiac event; PCI, percutaneous coronary intervention; TyG, triglyceride–glucose index.

**Figure 4 nutrients-15-02808-f004:**
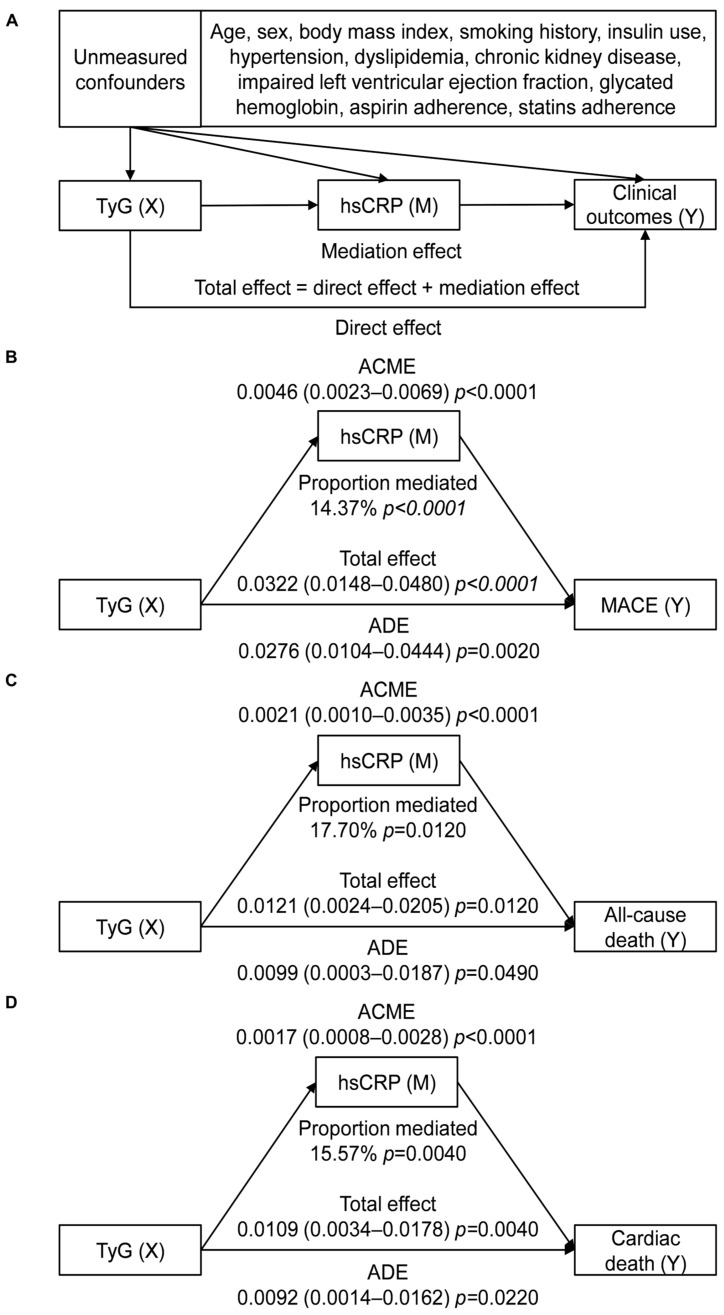
Mediation effect of hsCRP on the association between TyG and cardiovascular events. (**A**) Directed acyclic graph; (**B**) mediation analysis for MACE; (**C**) mediation analysis for all-cause death; (**D**) mediation analysis for cardiac death. Abbreviations: ACME, average causal mediation effect; ADE, average direct effect; hsCRP, high-sensitivity C-reactive protein; MACE, major adverse cardiac event; TyG, triglyceride–glucose index.

**Table 1 nutrients-15-02808-t001:** Baseline characteristics of patients grouped by TyG and hsCRP levels.

Variables	All Participants (*n* = 4419)	L-TyG/L-hsCRP (*n* = 602)	L-TyG/H-hsCRP (*n* = 206)	H-TyG/L-hsCRP (*n* = 2153)	H-TyG/H-hsCRP (*n* = 1458)	*p*
Demographic characteristics						
Age, years	62 (55–68)	63 (56–69)	63 (55–69)	61 (55–68)	62 (55–68)	0.005
≥65	1640 (37.11)	250 (41.53)	81 (39.32)	754 (35.02)	555 (38.07)	0.018
Female	1228 (27.79)	135 (22.43)	49 (23.79)	595 (27.64)	449 (30.80)	<0.001
Body mass index, kg/m^2^	26.04 (24.22–28.28)	24.86 (22.86–26.82)	25.81 (23.86–27.73)	25.96 (24.22–28.08)	26.73 (24.77–29.05)	<0.001
≥28	1212 (27.43)	90(14.95)	44 (21.36)	567 (26.34)	511 (35.05)	<0.001
Smoking history						<0.001
Current smoker	730 (16.52)	52 (8.64)	26 (12.62)	354 (16.44)	298 (20.44)	
Former smoker	1736 (39.28)	256 (42.52)	95 (46.12)	833 (38.69)	552 (37.86)	
Non-Smoker	1953 (44.20)	294 (48.84)	85 (41.26)	966 (44.87)	608 (41.70)	
Clinical characteristics						
Insulin use	635 (14.37)	68 (11.30)	26 (12.62)	313 (14.54)	228 (15.64)	0.069
Hypertension	4133 (93.53)	561 (93.19)	188 (91.26)	2013 (93.50)	1371 (94.03)	0.479
Dyslipidemia	4272 (96.67)	570 (94.68)	199 (96.60)	2077 (96.47)	1426 (97.81)	0.004
Peripheral artery disease	334 (7.56)	44 (7.31)	19 (9.22)	156 (7.25)	115 (7.89)	0.706
Chronic kidney disease	160 (3.62)	24 (3.99)	8 (3.88)	65 (3.02)	63 (4.32)	0.208
COPD	61 (1.38)	10 (1.66)	5 (2.43)	26 (1.21)	20 (1.37)	0.479
Prior myocardial infarction	951 (21.52)	126 (20.93)	30 (14.56)	470 (21.83)	325 (22.29)	0.083
Prior stroke	788 (17.83)	110 (18.27)	46 (22.33)	364 (16.91)	268 (18.38)	0.214
Prior revascularization	1531 (34.65)	227 (37.71)	54 (26.21)	765 (35.53)	485 (33.26)	0.012
Laboratory tests						
FBG, mmol/L	6.69 (5.58–8.24)	5.32 (4.67–6.10)	5.34 (4.70–6.19)	6.99 (5.91–8.54)	7.16 (5.98–8.86)	<0.001
FBG, mg/dL	120.42 (100.44–148.32)	95.76 (84.06–109.80)	96.12 (84.60–111.42)	125.82 (106.38–153.72)	128.88 (107.64–159.48)	<0.001
HbA1c, %	6.90 (6.30–7.90)	6.30 (5.80–6.90)	6.50 (6.00–7.20)	7.00 (6.30–7.90)	7.20 (6.50–8.20)	<0.001
HbA1c, mmol/mol	52 (45–63)	45 (40–52)	48 (42–55)	53 (46–63)	55 (48–66)	<0.001
LDL-c, mmol/L	2.16 (1.72–2.80)	1.80 (1.49–2.24)	1.92 (1.56–2.54)	2.14 (1.74–2.76)	2.42 (1.91–3.10)	<0.001
≤1.8	1302 (29.46)	307 (51.00)	89 (43.20)	619 (28.75)	287 (19.68)	<0.001
Triglyceride, mmol/L	1.42 (1.06–1.98)	0.84 (0.71–0.98)	0.87 (0.75–1.00)	1.56 (1.22–2.07)	1.66 (1.29–2.28)	<0.001
TyG	8.96 (8.59–9.38)	8.22 (8.04–8.34)	8.25 (8.12–8.36)	9.06 (8.77–9.43)	9.17 (8.83–9.55)	<0.001
hsCRP, mg/L	1.46 (0.70–2.92)	0.66 (0.32–1.18)	3.71 (2.67–7.87)	0.91 (0.49–1.39)	3.79 (2.62–6.39)	<0.001
LVEF < 40%	93 (2.10)	13 (2.16)	6 (2.91)	38 (1.76)	36 (2.47)	0.422
Lesion characteristics						
LM/TVD	2202 (49.83)	264 (43.85)	111 (53.88)	1036 (48.12)	791 (54.25)	<0.001
SYNTAX score						0.020
≤22	3740 (84.67)	530 (88.04)	171 (83.01)	1840 (85.50)	1199 (82.29)	
23–32	557 (12.61)	56 (9.30)	27 (13.11)	259 (12.04)	215 (14.76)	
≥33	120 (2.72)	16 (2.66)	8 (3.88)	53 (2.46)	43 (2.95)	
PCI status						0.024
Successful PCI	2712 (61.37)	339 (56.31)	121 (58.74)	1323 (61.45)	929 (63.72)	
Unsuccessful PCI	129 (2.92)	13 (2.16)	5 (2.43)	69 (3.20)	42 (2.88)	
No PCI	1578 (35.71)	250 (41.53)	80 (38.83)	761 (35.35)	487 (33.40)	
Medication adherence						
Aspirin						0.181
2-year regular	3276 (74.13)	452 (75.08)	155 (75.24)	1612 (74.87)	1057 (72.50)	
1-year regular	926 (20.95)	126 (20.93)	35 (16.99)	441 (20.48)	324 (22.22)	
Irregular/<1 year	217 (4.91)	24 (3.99)	16 (7.77)	100 (4.64)	77 (5.28)	
Statins						0.217
2-year regular	2743 (62.07)	386 (64.12)	126 (61.17)	1349 (62.66)	882 (60.49)	
1-year regular	1161 (26.27)	138 (22.92)	55 (26.70)	576 (26.75)	392 (26.89)	
Irregular/<1 year	515 (11.65)	78 (12.96)	25 (12.14)	228 (10.59)	184 (12.62)	

Values are presented as number (%) or median (interquartile range). Abbreviations: CAD, coronary artery disease; COPD, chronic obstructive pulmonary disease; FBG, fasting blood glucose; H, high; HbA1c, glycated hemoglobin; hsCRP, high-sensitivity C-reactive protein; L, low; LDL-c, low-density lipoprotein cholesterol; LM/TVD, left main stem/three-vessel disease; LVEF, left ventricular ejection fraction; PCI, percutaneous coronary intervention; SYNTAX, synergy between percutaneous coronary intervention with Taxus and cardiac surgery; TyG, triglyceride–glucose index.

**Table 2 nutrients-15-02808-t002:** Association of TyG and hsCRP with cardiovascular events.

Clinical Outcome	Group	No. of Events (%)	Event Rate per 1000 pys	Crude Model		Adjusted Model	
				HR (95% CI)	*p*	HR (95% CI)	*p*
MACE	L-TyG/L-hsCRP	32 (5.32)	25.11	Reference		1.0	
	L-TyG/H-hsCRP	18 (8.74)	42.74	1.70 (0.95, 3.02)	0.073	1.46 (0.82, 2.62)	0.198
	H-TyG/L-hsCRP	203 (9.43)	45.40	1.81 (1.24, 2.62)	0.002	1.78 (1.22, 2.60)	0.003
	H-TyG/H-hsCRP	152 (10.43)	49.63	1.97 (1.34. 2.88)	0.001	1.83 (1.24, 2.70)	0.002
	*p* for trend			0.001		0.003	
All-cause death	L-TyG/L-hsCRP	5 (0.83)	3.81	Reference		Reference	
	L-TyG/H-hsCRP	4 (1.94)	9.05	2.38 (0.64, 8.88)	0.195	1.42 (0.37, 5.39)	0.607
	H-TyG/L-hsCRP	37 (1.72)	7.87	2.06 (0.81, 5.24)	0.129	2.89 (1.10, 7.56)	0.031
	H-TyG/H-hsCRP	39 (2.67)	12.11	3.16 (1.25, 8.03)	0.015	3.96 (1.51, 10.36)	0.005
	*p* for trend			0.009		0.001	
Cardiac death	L-TyG/L-hsCRP	2 (0.33)	1.53	Reference		Reference	
	L-TyG/H-hsCRP	3 (1.46)	6.79	4.43 (0.74, 26.51)	0.103	2.64 (0.42, 16.44)	0.298
	H-TyG/L-hsCRP	22 (1.02)	4.68	3.10 (0.73, 13.19)	0.126	4.23 (0.94, 19.02)	0.061
	H-TyG/H-hsCRP	24 (1.65)	7.45	5.02 (1.18, 21.31)	0.029	5.94 (1.32, 26.79)	0.021
	*p* for trend			0.018		0.007	
Myocardial infarction	L-TyG/L-hsCRP	5 (0.83)	3.83	Reference		Reference	
	L-TyG/H-hsCRP	5 (2.43)	11.49	2.97 (0.86, 10.24)	0.086	2.67 (0.77, 9.26)	0.122
	H-TyG/L-hsCRP	56 (2.60)	12.16	3.15 (1.26, 7.87)	0.014	3.22 (1.28, 8.11)	0.013
	H-TyG/H-hsCRP	52 (3.57)	16.43	4.28 (1.71, 10.71)	0.002	4.00 (1.58, 10.17)	0.004
	*p* for trend			0.001		0.002	
Any revascularization	L-TyG/L-hsCRP	27 (4.49)	21.11	Reference		Reference	
	L-TyG/H-hsCRP	11 (5.34)	25.70	1.22 (0.61, 2.46)	0.575	1.11 (0.55, 2.25)	0.768
	H-TyG/L-hsCRP	140 (6.50)	30.86	1.46 (0.97, 2.20)	0.072	1.35 (0.89, 2.04)	0.159
	H-TyG/H-hsCRP	104 (7.13)	33.50	1.57 (1.03, 2.40)	0.036	1.40 (0.91, 2.17)	0.127
	*p* for trend			0.033		0.116	

Adjusted for age (continuous), sex, body mass index (continuous), smoking history, insulin use, peripheral artery disease, left ventricular ejection fraction <40%, left main stem/three-vessel disease, synergy between percutaneous coronary intervention with Taxus and cardiac surgery score (categorical), percutaneous coronary intervention status, aspirin adherence, statin adherence. Abbreviations: CI, confidence interval; H, high; HR, hazard ratio; hsCRP, high-sensitivity C-reactive protein; L, low; MACE, major adverse cardiac event; No., number; TyG, triglyceride–glucose index.

## Data Availability

The datasets generated and analyzed during the current study are available from the corresponding author upon reasonable request.

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
