# Peer review of "Inflammation and Insulin Resistance in Diabetic Chronic Coronary Syndrome Patients"

_nutrients, 2023, doi:10.3390/nu15122808_

Round 1
Reviewer 1 Report
Please read the attachment. Thank you.

Technical English appears to be of a good standard, effectively communicating the study objectives, methods, and findings.
Author Response
Dear Reviewer,
Thank you for your valuable feedback. We have made additional revisions to address the comments of a new reviewer. Please refer to the updated version of our response.

Reviewer 2 Report
Congratulations for the study. The limiting point of this article is the discussion. An interpretation of the results found is expected in this section of the article. Specifically in this study in which the correlation between risk factors and clinical outcome is the focus, it is expected that the discussion will bring a more robust interpretation of the findings. However, it is observed in the second, third and fourth paragraphs findings from other studies, referring the reader to a new introduction. Given the relevance of the study, I recommend that the discussion be redone with a more expressive focus on the results of the study, valuing the data presented.
Author Response
Thank you for your time and effort in reviewing our manuscript. We also appreciate your positive feedback on our work. Based on your comments, we have thoroughly revised the discussion section to provide a more comprehensive and focused interpretation of the findings.
Additionally, we have made extensive modifications to the manuscript and supplementary materials in response to all reviewers' comments. These include simplifying the title, rewriting the Introduction and Discussion sections, adjusting for adherence to aspirin and statins in the re-analysis, relocating the study flowchart to the main text, and moving a previous table to the supplementary materials. Furthermore, we have conducted an additional sensitivity analysis to assess the impact of removing imputed data. Several minor changes, such as adding new references, providing additional methodological details, and expanding the discussion of limitations, have also been implemented.
We greatly appreciate your invaluable assistance in improving the quality of our work. If you have any further suggestions or require additional modifications, we would be more than happy to address them.
Reviewer 3 Report
This paper provides information about the effects of systemic inflammation on the association between insulin resistance and cardiovascular events in diabetic patients with chronic coronary syndrome. This research article reports that controlling inflammation in insulin-resistant patients may reduce major adverse cardiac events, bringing new contributions to the field. However, I have some suggestions and concerns about the paper described below.
Introduction
- To highlight the importance of the study carried out by the authors, please include the epidemiological data from patients with diabetes and chronic coronary syndrome, demonstrating how the world population is affected by these conditions.
- Please add a paragraph describing how insulin resistance present in diabetic patients acts as a cardiovascular risk factor.
- Please add a paragraph relating diabetes and inflammation, as well as their cardiovascular implications.
Materials and Methods
- Please remove the aim of the study from the methodology, lines 90 to 92.
- The authors need better characterize the inclusion and exclusion criteria as described in the supplementary material and should cite the supplementary material referring to these criteria in the text.
- The authors need to describe in the study design and participants' topic the total number of patients used after applying the exclusion criteria and the sex of the patients in the study. Moreover, how did the authors divide the patients to perform the analyses? Each group that appears in Table 1 should describe in detail.
Reviewer 4 Report
Line 145 – 146: the authors should clarify in the text how they assessed normality.
“When analyzed as categorical variables, TyG was classified into low (L)-TyG and high (H)-TyG groups at the optimal cut-off value for predicting MACE determined by the receiver operating characteristic analysis, and hsCRP was classified into L-hsCRP and H-hsCRP groups at 2.00 mg/L according to the definition of residual inflammatory risk.” Converting numeric variables to categoric has important costs and should generally be avoided (see, e.g. https://www.ncbi.nlm.nih.gov/pmc/articles/PMC1458573/).
The authors state that they also used restricted cubic splines with 4 knots but this was only limited to a single sensitivity analysis reported in the supplementary electronic materials. It would have been preferrable to have the categorical variable analysis as a supplementary material and the main analysis with the continuous variables. The authors should also clarify how they chose the number of knots.
Lines 156 – 159: Apparently all patients used aspirin and statins, but different INNs for statins and different doses for both could be used. Therefore, medication use should have also been a covariate, especially considering that both aspirin and statins are known to exert anti-inflammatory effects. Same for lines 173 – 176. Absence of any adjustment for aspirin and statins should be acknowledged as a limitation in the discussion.
Line 178: “Statistical analyses were conducted with R version 4.2.0 (R Core Team 2022, Vienna, Austria)”. The authors should clarify whether additional R packages besides “mediation” were used within the analyses (for instance what package was used to determine ROC curves and optimal cut-off values?).
Lines 184 – 185: Only 27.79% of the study population were female, which seems far from a representative sample of the general population (where one would expect a quasi-equal proportion between males and females). What is the explanation for this sex imbalance? (this should be part of the Discussion section).
Sensitivity analyses should also have been conducted for the imputation (i.e. comparing results on the dataset without imputed values with results on the imputed dataset).
